

# Minor contributions of daytime monoterpenes are major contributors to atmospheric reactivity

Deborah F. McGlynn[1], Graham Frazier[1], Laura E.R. Barry[2], Manuel T. Lerdau[2,3], Sally E. Pusede[2], and Gabriel Isaacman-VanWertz[1]

[1]Department of Civil and Environmental Engineering, Virginia Tech, Blacksburg, VA, 24061, USA
[2]Department of Environmental Sciences, University of Virginia, Charlottesville, VA, 22904, USA
[3]Department of Biology, University of Virginia, Charlottesville, VA, 22904, USA

**Correspondence:** Gabriel Isaacman-VanWertz (ivw@vt.edu)

## 1 Abstract

Emissions from natural sources are driven by various external stimuli such as sunlight, temperature, and soil moisture. Once biogenic volatile organic compounds (BVOCs) are emitted into the atmosphere, they rapidly react with atmospheric oxidants, which has significant impacts on ozone and aerosol budgets. However, diurnal, seasonal, and interannual variability of these

species are poorly captured in emissions models due to a lack of long-term, chemically speciated measurements. Therefore, increasing the monitoring of these emissions will improve the modeling of ozone and secondary organic aerosol concentrations. Using two years of speciated hourly BVOC data collected at the Virginia Forest Lab (VFL), in Fluvanna County, Virginia, we examine how minor changes in the composition of monoterpenes between seasons are found to have profound impacts on ozone and OH reactivity. The concentration of a range of BVOCs in the summer were found to have two different diurnal

profiles, which we demonstrate appear to be driven by light-dependent versus -independent emissions. Factor analysis was used to separate the two observed diurnal profiles and determine the contribution from each driver. Highly reactive BVOCs were found to have a large influence on atmospheric reactivity in the summer, particularly during the daytime. These findings reveal a need to monitor species with high atmospheric reactivity but have low concentrations and to more accurately capture their emission trends in models.

## 2 Introduction

Biogenically emitted volatile organic compounds (BVOCs) are important precursors for reactions with atmospheric oxidants and secondary organic aerosol (SOA) formation (Atkinson and Arey, 2003a; Guenther et al., 1995, 2000; Kroll and Seinfeld, 2008). Their emissions are primarily driven by the species of plants present and by changes in temperature and light, with secondary effects of other ecological factors. Light dependent or *de novo* biosynthesis emissions are produced within the leaves

of plants and emitted shortly after formation through plant stomata (Niinemets and Monson, 2013). These emissions tend to increase with temperature (Guenther et al., 2006; Guenther, 1997) but also require light. The dominant *de novo* BVOC emitted



is isoprene, though some monoterpenes can be emitted in this manner (Staudt and Seufert, 1995; Tingey et al., 1979; Ghirardo et al., 2010; Taipale et al., 2011). In contrast, other emissions occur independently of light from a wide variety of vegetation and therefore occur year-round primarily with a temperature dependence (Niinemets and Monson, 2013; Ghirardo et al., 2010; Guenther et al., 1991). Monoterpenes, sesquiterpenes, and diterpenes are largely emitted in a temperature dependent manner through volatilization from storage pools or resin ducts from within the plant (Zimmerman, 1979; Niinemets and Monson, 2013; Lerdau et al., 1997; Lerdau and Gray, 2003). The rate of volatilization is determined by the compound's vapor pressure (Lerdau and Gray, 2003).

The diurnal concentration profile of individual species (i.e., the observed average variability within a 24-hour period) is a function of the drivers of emissions, the concentrations of atmospheric oxidants, and meteorology. For isoprene, which is emitted from plants in a light-dependent manner (Niinemets and Monson, 2013; Ghirardo et al., 2010; Guenther et al., 1991; Bouvier-Brown et al., 2009), the diurnal profile is well established and relatively consistent across environments (Rinne et al., 2002; Guenther et al., 2000; Delwiche and Sharkey, 1993; Niinemets and Monson, 2013). Due to strong daytime emissions, concentrations peak midday to late afternoon, when incoming solar radiation and temperatures are greatest. Nighttime emissions of *de novo* emitted BVOCs drop to near zero due to the lack of light (Niinemets and Monson, 2013; Ghirardo et al., 2010; Panopoulou et al., 2020; Guenther et al., 1996; Rinne et al., 2002). Concentrations of *de novo* emitted species concomitantly drop as suspended gases are depleted by atmospheric oxidation.

The diurnal variation of monoterpenes is substantially more variable and complex. Because their emissions are predominantly temperature dependent, emissions peak in the afternoon but continue throughout the night. Consequently, monoterpene concentrations are often greatest during the evening hours (Bouvier-Brown et al., 2009; Panopoulou et al., 2020; Hakola et al., 2012), when oxidation by photochemically formed hydroxyl radicals is minimal and boundary height is reduced, decreasing dilution through atmospheric mixing (Panopoulou et al., 2020; Bouvier-Brown et al., 2009; Haapanala et al., 2007). However, some plants do produce and emit monoterpenes in a light-dependent manner (Staudt et al., 1999; Staudt and Seufert, 1995; Harley et al., 2014; Yu et al., 2017; Taipale et al., 2011; Guenther et al., 2012). Despite these findings, light dependent monoterpene emission have largely been deemed to contribute minimally to total monoterpene emissions. (Bouvier-Brown et al., 2009; Lerdau and Gray, 2003). This lack of contribution to total flux occurs because they are emitted from only a handful of plant taxa and the emission rates themselves have not been shown to be significant (Staudt et al., 1999; Loreto et al., 1998; Staudt and Seufert, 1995). Interestingly, a few studies find that many trees emit low levels of monoterpenes in a light dependent manner, and these studies have found that this emission activity is seasonal and changes with phenological patterns. (Fischbach et al., 2002; Ghirardo et al., 2010) (Ghirardo et al., 2010; Taipale et al., 2011; Steinbrecher et al., 1999). Despite representation of light dependent and independent monoterpene emissions, discrepancies exist between this representation and the literature (Guenther et al., 2012; Tingey et al., 1979; Rinne et al., 2002; Steinbrecher et al., 1999; Bouvier-Brown et al., 2009; Taipale et al., 2011; Ghirardo et al., 2010; Kesselmeier and Staudt, 1999; Fischbach et al., 2002; Staudt and Seufert, 1995).

The explanation for these discrepancies among studies appears to lie in the fact that for some plant species, e.g., members of the genus Pinus, monoterpene emissions are largely not light dependent, though this also tends to vary with season (Kesselmeier and Staudt, 1999; Harley et al., 2014; Niinemets et al., 2002). While for other plants species, e.g., Fagus and the European live





oaks (sub-genus Cerris), emissions are largely light dependent (Schuh et al., 1997; Niinemets and Monson, 2013). The observed variability appears to be a function of both plant species, terpene species, and possibly the plant ecosystem (Ghirardo et al., 2010; Staudt and Seufert, 1995; Niinemets et al., 2002; Steinbrecher et al., 1999). That is, the same terpenoid compound may

be light dependent in one species but light independent in another. From the perspective of atmospheric processes, though, the impacts of monoterpenes depend on their absolute fluxes, the timing and control over these fluxes, and their specific reactivities. A major goal of the present work is to understand the potential role that the minor contribution of light dependent emissions and/or individual compounds with differing temporal variability may play in the atmosphere. Certain monoterpenes that are often emitted at low levels and/or in a light dependent manner have extremely high reactivities, raising the question of whether

or not chemical impact may be disproportionate to flux magnitude.

A lack of understanding of how individual compounds are emitted from vegetative sources makes emission modeling difficult and more uncertain. This is largely due to the impact the structure of a BVOC has on its aerosol formation potential and its reaction rates with atmospheric oxidants, particularly for reactions involving ozone. For example, endocyclic monoterpenes (e.g., limonene and 3-carene) and sesquiterpenes (e.g., $\alpha$-humulene and $\beta$-caryophyllene) have a greater aerosol formation

potential and tend to react faster than compounds with exocyclic double bonds (e.g. $\alpha$-pinene, $\alpha$-cedrene). Consequently, long-term measurements of speciated BVOCs can assist in modeling BVOC emissions and in understanding their contribution to ozone modulation and SOA formation (Porter et al., 2017). These impacts extend further to the importance of individual fast-reacting isomers, which can represent substantial fractions of total reactivity even at low concentrations (Yee et al., 2018). In this context, a detailed understanding of the different drivers of isomer emissions and the temporal variability of composition

is critical for interpreting such data.

Using two years of chemically resolved concentration measurements of in-canopy, biogenic volatile organic compound (BVOC) concentration data, we examine the contribution of individual monoterpene compounds to ozone reactivity on diurnal, seasonal, and interannual timescales. We elucidate the impact of temporal variability on ozone reactivity on scales from hours to years by identifying two varying components in the data, which we identify as coming from light dependent and

independent emissions and quantifying their chemical impacts on each timescale. Factor analysis is used to quantitatively separate these observed profiles and their contributions to total monoterpene concentration and ozone and OH reactivity. Our findings highlight the need to better understand the drivers of emissions with isomer-level chemical resolution and improve their representation in emissions models as they have significant atmospheric impact.

## 3  Methods

### 3.1  Data collection and preparation

We measured in-canopy BVOC concentrations at the Virginial Forest Lab (VFL, 37.9229 °N, 78.2739 °W) in Fluvanna County, Virginia. The VFL sits on the east side of the Blue Ridge Mountains and is about 25 km east-southeast of Charlottesville, VA. The site houses a 40-meter meteorological tower, with a climate-controlled, internet-connected lab at the bottom that is supplied by line power. The BVOC concentrations were measured using a gas chromatography flame ionization detector (GC-





FID) adapted for automated collection and analysis of air samples from mid-canopy (∼20 m) of the VFL. Additional details

pertaining to the measurement location, instrument operation, and data analyses can be seen in (McGlynn et al., 2021). To

identify analytes in the samples, a mass spectrometer (MS, Agilent 5977) was deployed in October 2019, September 2020, and

June 2021 in parallel with the FID. Retention times of analytes detected by the two detectors were aligned using the retention

time of known analytes. Analytes were identified by mass spectral matching with the 2011 NIST MS Library and reported

retention indices (Mass Spectrometry Data Center, NIST, 2022). The chromatographic data were analysed using the freely-

available TERN software packaged by (Isaacman-VanWertz et al., 2017) within the Igor Pro 8 programming environment

(Wavemetrics, Inc.). The measurement period included in this work extends from September 15, 2019, to September 14, 2021.

This work presents all isoprene and monoterpene data collected during the measurement period but focuses largely on the

monoterpenes between the months of May-September.

## 3.2    Positive Matrix Factorization

Positive matrix factorization (PMF) has been widely used for source apportionment problems (Norris et al., 2014; Ulbrich et al.,

2009; Kuang et al., 2015). A large number of variables can be reduced by the PMF algorithm to the main sources or factors that

drive the observed variability (Norris et al., 2014). Application of PMF to multi-variable data generates two matrices, the factor

contributions and factor profiles (Norris et al., 2014), which for environmental data represent timeseries as a set of covarying

variables (e.g., chemical species).

This work employed the United States Environmental Protection Agency's (EPA) PMF 5.0 program to support the identifi-

cation in the observational data of two apparent sources or drivers of BVOC concentration variability. Specifically, a two-factor

PMF solution was examined to better understand and quantify the profiles and temporal variability of each observed factor.

The two years of monoterpene data were run separately ("2020": September 15[th], 2019-September 14[th], 2020, and "2021":

September 15[th], 2020- September 14[th], 2021), with uncertainty, u, in the data calculated using the equation provided by (Norris

et al., 2014):

$$u = \sqrt{(0.15 \times concentration)^2 + (0.5 \times MDL)^2} \tag{1}$$

The method detection limit, MDL, is 2.2 ppt for monoterpenes (McGlynn et al., 2021). Values below the method detection

limit were substituted with MDL/2 in both the concentration and uncertainty file. Missing data are excluded from the data

processing (Norris et al., 2014). Factor contributions are returned from the PMF program as normalized values, which are

converted to concentration by multiplying returned values by the sum of the concentrations of species in the factor profiles.

## 3.3    Reactivity calculations

Reactivity of an individual BVOC with ozone (O3R) and OH (OHR) is calculated as the sum of the products of the concentra-

tion and oxidation reaction rate constant of each BVOC,i:



$$OxR_{tot}(s^{-1}) = \sum (k_{O_x + BVOC_i}[BVOC_i]) \qquad (2)$$

All rate constants (units: cm$^3$ molec$^{-1}$ s$^{-1}$) used in this work listed in Table S1 (Atkinson et al., 2006, 1990a; Pinto et al., 2007; Atkinson and Arey, 2003b; Shu and Atkinson, 1994; Pratt et al., 2012; Atkinson et al., 1990b). A temperature of 298 K is assumed for all rate constants, representing the approximate midpoint between day and night temperatures in the summer at this site, which vary by roughly 10°C (McGlynn et al., 2021). Taking the temperature dependence of rate constants into

account would increase daytime OH reactivity by 5-8%, and decrease nighttime OH reactivity by approximately the same amount (National Institute for Standards and Technology, 2019). These differences suggest the true difference between the light dependent (daytime) and light independent (nighttime) mixtures is ~10% higher than calculated, but this effect is not included quantitatively because temperature dependence is not known for many monoterpene reaction rates.

## 4   Results and discussion

At the VFL, concentrations of a wide range of species, including anthropogenic and other VOCs, are measured hourly. The BVOCs measured include isoprene, methyl vinyl ketone, methacrolein, 11 monoterpenes, and 2 sesquiterpenes. This work focuses primarily on monoterpenes, which contribute the large fraction of speciated ozone and OH reactivity from BVOCs (McGlynn et al., 2021) at the research site throughout the year.

### 4.1   Monoterpene seasonality

To understand the drivers of monoterpene variability, we first examine diurnal and seasonal patterns in two monoterpenes found at the site, $\alpha$-pinene and limonene, that exhibit features of two different concentration profiles. Seasonal averages are defined as: December, January, and February (Winter); March, April, May (Spring); June, July, August (Summer); and September, October, November (Fall). Diurnal trends in these species demonstrate some clear differences in their concentration patterns (Figure 1). $\alpha$-pinene concentrations were lowest in the daytime winter hours at about 0.05 ppb and highest in the evening

summer hours, at 0.60 ppb. In all seasons, $\alpha$-pinene concentrations were highest at night and decreased in the morning hours, following "typical" patterns of temperature-driven monoterpene concentrations (Bouvier-Brown et al., 2009) due to the higher planetary boundary layer and increased concentrations of oxidants during the day. Concentrations were lowest in the middle of the day, between 10:00 and 17:00 and highest between 20:00 and 8:00 (Figure 1a). Concentration transitions between these periods vary somewhat by season in accordance with the changing temperature and daylight hours of a subtropical climate

zone.

In contrast, while limonene concentrations were similarly lowest in the daytime winter hours, at 0.01 ppb, they were highest during the daytime summer hours, at 0.2 ppb. In fall, winter, and spring, limonene exhibited the same seasonality as $\alpha$-pinene with daytime lows and nighttime highs, though with weaker diurnal variability (Figure 1b). In summer, however, diurnal trends in limonene concentrations are very different, with a peak in the mid to late afternoon. To reach daytime peaks in concentration,

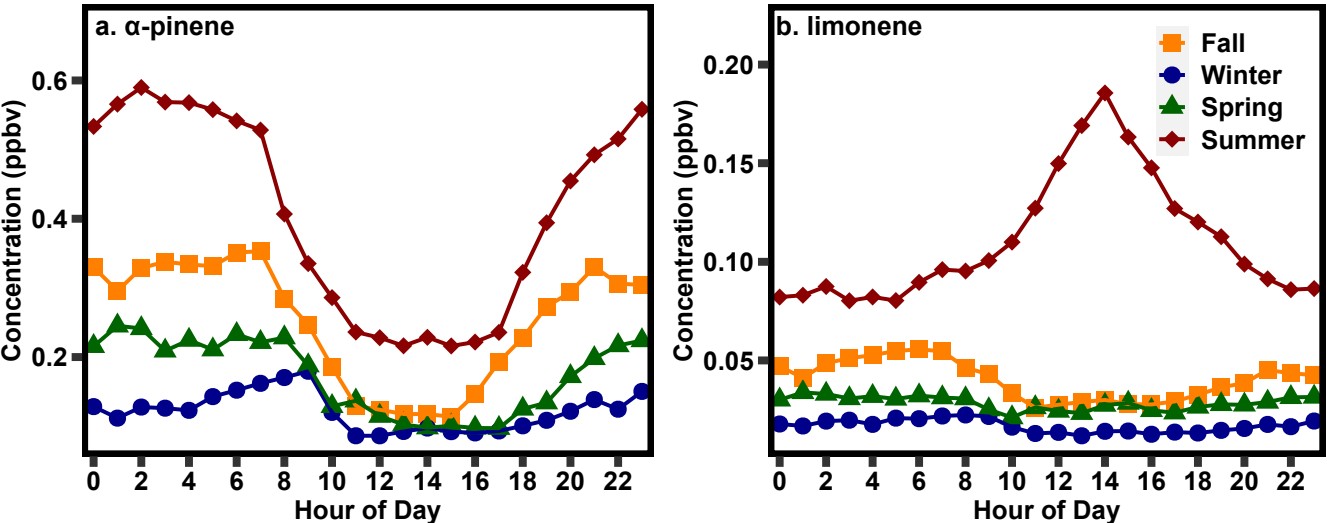

**Figure 1.** The mean (a) $\alpha$-pinene and (b) limonene concentration in the four seasons of the northern hemisphere between September 2019 and September 2021.

daytime emissions of limonene must be high, particularly given that the reaction rate of limonene with OH radical, and ozone, is 3, and 2.3, times as fast, respectively, as those of $\alpha$-pinene.

    The seasonal rise and fall in the observed daytime peak of limonene, in contrast to the relative stability of $\alpha$-pinene, is apparent in a spring/summertime comparison of daytime (7AM - 7PM) and night-time (7PM - 7AM) average concentrations (Figure 2). The full two-year time series of this plot can be found in the supplemental document (Figure S1). As observed
in the diurnal profiles, $\alpha$-pinene evening concentrations are higher than daytime concentrations throughout the year; while concentrations increase in the summer, this increase is observed in both daytime and nighttime concentrations (Figure 2a). In contrast, while concentrations of limonene are highest at night throughout the early spring, concentrations begin to peak in the daytime in late-May (Figure 2b). From late-May through mid-September, concentrations are highest during the day, suggesting a strong daytime source of limonene specifically in the summer, which may be co-emitted with other monoterpenes
but is not a strong feature for $\alpha$-pinene. The daytime peak in limonene is unique to summer and occurs in both years (Figures 1, 2, and S1). Interestingly, while the daytime peak in summer is relatively consistent across years, nighttime concentrations of limonene in the summer are substantially lower in 2021 compared to 2020 (Figure S1), suggesting sources for daytime and nighttime limonene that differ in their interannual variation. However, additional years of data are likely necessary to better understand the driver of this interannual variability. We demonstrate below that the timing of the rise and fall of the strong
daytime source of limonene correlates with concentrations of isoprene, a known *de novo* emitted BVOC species, and appears to be a component of a set of light-dependent monoterpene emissions.





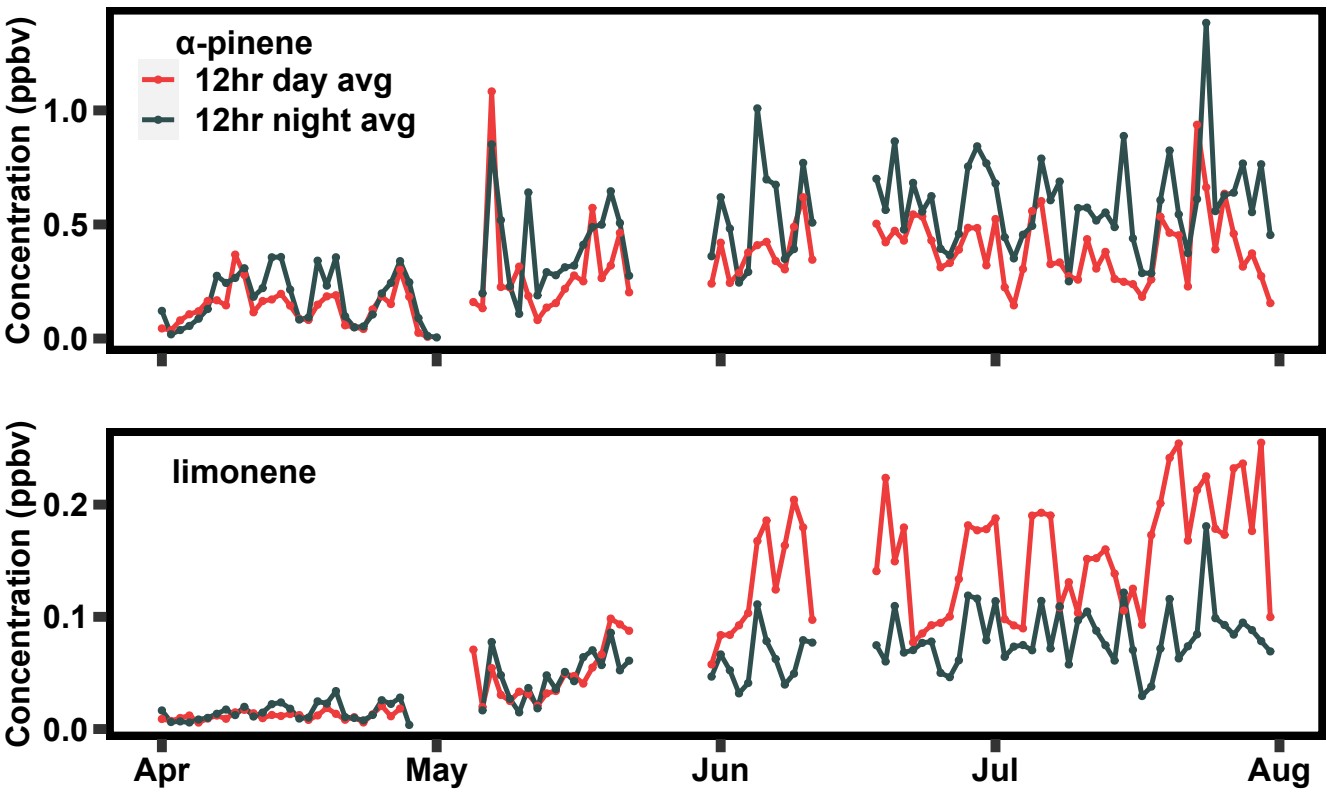

**Figure 2.** The 12-hour average of $\alpha$-pinene and limonene between April 2021 and August 2021. The averaging period for each compound was between 7 AM and 7 PM.

## 4.2 Light dependent and light independent monoterpene concentration

To better characterize the observed light-dependent monoterpenes and quantify their impacts, the patterns in monoterpenes were deconvolved as two factors using PMF. The determined factors demonstrate a clear separation between a set of monoterpenes that exhibit only nighttime peaks in concentration, and a set of compounds that exhibit a tendency to have high daytime concentrations. Quantitative assessment of the uncertainty of the two-factor solution is performed using bootstrapping, in which 100 runs are performed using arbitrary subsets of data; 95% of bootstrap runs reproduce both factors (Table S2) with no unmapped base factors. An unmapped base factor indicates that one or more bootstrap runs did not correlate with a determined factor from the base model run (Norris et al., 2014). The Pearson correlation coefficient threshold used for this analysis was the EPA PMF default value of 0.6 (Norris et al., 2014).

A "light dependent" factor is present primarily during the summer, characterized by daytime peaks that roughly coincide with the seasonality and variability of isoprene (Figure 3, results from 2020 shown, results from 2021 in Figure S1). This factor even mirrors transient decreases in concentrations observed in isoprene, such as those observed in June 2020, July

**Figure 3.** Time series of isoprene concentration, the two positive matrix factorization factors between September 2019 and September 2020 and the breakdown of the monoterpene species that contribute to each factor.

2020, and September 2020, denoted with black arrows in Figure 3a, b. The largest contributor to the light dependent factor is

limonene (roughly one-third), followed by cymene, sabinene, and a relatively small contribution from $\alpha$-pinene, denoted by the pie charts above each factor time series. A table indicating the percent contribution for the species in each factor can be found in Table S1. A more dominant factor contains most of the $\alpha$- and $\beta$-pinene and exhibits a diurnal pattern and seasonality more in line with what is typical for temperature-driven monoterpenes; this factor is referred to as "light independent" to distinguish it and because the dominant biogenic emission model (MEGAN) distinguishes between emission pathways as light

dependent (i.e., *de novo*) vs. independent (i.e., temperature-driven volatilization from storage pools) (Guenther et al., 2012). Interpretation of factors is further supported by their diurnal trends, a representative sample of which is shown in Figure 4. The light dependent factor peaks mid-day, following a similar temporal pattern as isoprene. We infer these monoterpenes to be emitted through similar processes as isoprene and attribute them to *de novo* emissions. In contrast, the higher-concentration

**Figure 4.** A four-day period in July 2020 of isoprene, and the two PMF factors (Light Dependent and Light Independent).

monoterpene factor peaks in the evening to early morning hours, following more typical monoterpene diurnal patterns. We

attribute these monoterpene concentrations to temperature-driven light independent emissions of monoterpenes.

Overall, the light dependent factor accounts for ∼25% of summertime monoterpene concentration, but at times the light dependent factor may contribute significantly or even dominate concentrations due to their differing diurnal variability in emissions. Interestingly, greater than 85% of the most dominant monoterpenes, including $\alpha$-pinene, $\beta$-pinene, tricyclene, fenchene, and camphene are found almost entirely in the light independent factor (Table 1). Conversely, greater than 85% of cymene,

sabinene, and thujene are found in the light dependent factor (Table 1). A small number of species are more split, with larger





**Table 1.** Percent of concentrations attributed to *de novo* and pool emissions by compound for 2019-2020

|  | **Annual** |  | **Summer** |  |
| --- | --- | --- | --- | --- |
| compound | % LIF | % LDF | % LIF | % LDF |
| $\alpha$-pinene | 97.7 | 2.3 | 96.6 | 3.4 |
| $\beta$-pinene | 96.1 | 3.9 | 94.2 | 5.8 |
| tricyclene | 94.3 | 5.7 | 91.8 | 8.2 |
| fenchene | 92.1 | 7.9 | 88.6 | 11.4 |
| camphene | 91.0 | 9.0 | 87.2 | 12.8 |
| $\beta$-phellandrene | 78.9 | 21.1 | 71.5 | 28.5 |
| $\gamma$-terpinene | 48.5 | 51.5 | 38.6 | 61.4 |
| limonene | 43.0 | 57.0 | 33.5 | 66.5 |
| thujene | 14.6 | 85.4 | 10.2 | 89.8 |
| cymene | 14.0 | 86.0 | 9.8 | 90.2 |
| sabinene | 0.0 | 100.0 | 0.0 | 100.0 |

percentages of their concentrations attributed to light dependent emissions than light independent emission in the summer months. These species include, $\beta$-phellandrene, limonene, and $\gamma$-terpinene (Table 1).

### 4.3 Ozone and OH reactivity

Despite the low contribution of the light dependent factor to total monoterpene concentration, this factor has a large impact on
ozone and OH reactivity. Comparing the stacked diurnal concentration profile (Fig. 5a) to the stacked ozone and OH reactivity diurnal profile (Fig. 5b, c) in summer illuminate's clear differences in their variability. While the concentration profile shows that the majority of species peak at night, there is a slight increase in the middle of the day, owing to the contribution from light dependent emissions. When this profile is multiplied by respective reaction rate constant for each species and oxidant, there is a clear mid-day peak that prevails as a significant contributor to ozone and OH reactivity in the summer. Further, the
largest contributor to total ozone and OH reactivity is limonene despite its lower contribution to total concentration due to its high reaction rate with each atmospheric oxidant.

A majority of the highly reactive isomer limonene is associated with light dependent monoterpenes (57%), while the more dominant $\alpha$-pinene concentrations are almost entirely attributed to pool emissions (98%, Table 1). Sabinene is also a notable contributor to the light dependent mixture, contributing approximately 30% to concentration, 25% to ozone reactivity, and
33% to OH reactivity; it is not found in the light independent mixture. The major contribution of limonene and sabinene to the light dependent monoterpene mixture makes light driven emissions particularly reactive, with a reaction rate roughly 1.5 times that of the light independent mixture for both ozone and OH reactivity. This daytime peak has an enormous impact on daytime ozone and OH reactivity (Fig. 5e, f), such that calculated summertime ozone and OH reactivity consequently have little diurnal pattern and is roughly uniform throughout the day (average: 1.4-2.4 $\times$ 10$^{-6}$ s$^{-1}$ for ozone reactivity and 1-2 s$^{-1}$





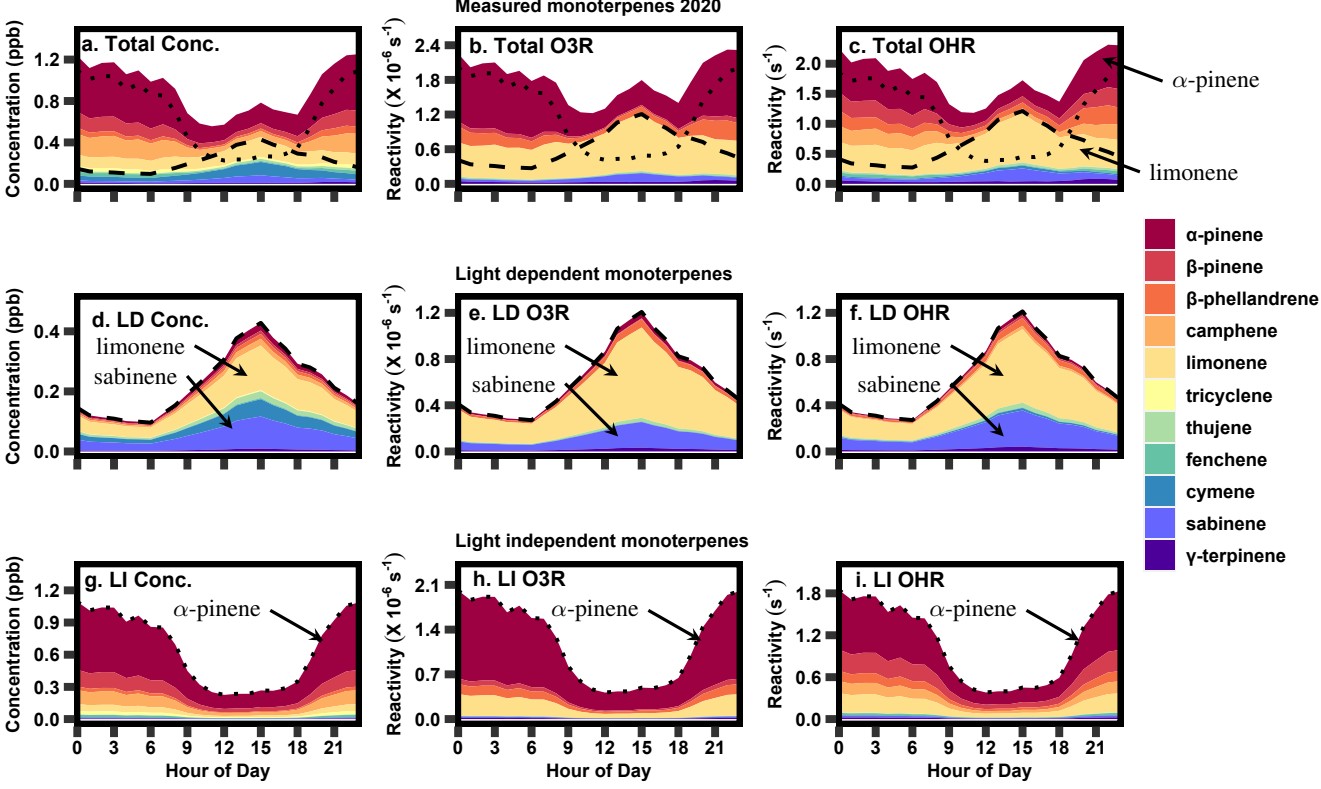

**Figure 5.** Time series of isoprene concentration, the two positive matrix factorization factors between September 2019 and September 2020 and the breakdown of the monoterpene species that contribute to each factor.

for OH reactivity) during the summer months. Even in the summer, when concentrations of light dependent monoterpenes are highest, the diurnal profile of the total monoterpene chemical class (Figure 5a) roughly follows that of $\alpha$-pinene (Figure 5a) with only moderate daytime concentrations. However, this average profile is a combination of a night-time peak dominated by light independent compounds (Figure 5g) and a daytime peak dominated by light dependent compounds (Figure 5d) that has a stronger contribution to reactivity. Consequently, understanding light dependent monoterpenes is critical, not only to better

characterize the carbon cycle and predict long-term trends, but also because it has immediate and substantial impacts on the atmospheric oxidant budget in the summer that would be overlooked when considering monoterpenes as a bulk compound class.

## 5   Conclusions

Using two years of hourly speciated BVOC concentrations collected at a meteorological tower in central Virginia, we identify

and quantify diurnal and seasonal variability of monoterpenes and isoprene. Though a majority of monoterpene concentrations





exhibit temporal behaviour expected from pool emissions whose flux rates are independent of light, we identify a minor (in mass terms) contribution from monoterpenes with seasonality and diurnal variability that show a strong light dependence and resemble *de novo* emissions. These light dependent monoterpene emissions are strongest in the summer, where they contribute ∼25% to total monoterpene concentrations, with smaller contributions in other seasons. However, the minor contribution to

total monoterpene mass belies their major impact on ozone and OH reactivity. Due to differences in the temporal variability of the two monoterpene classes and the significantly higher reaction rates of the light dependent mixture, we observe high ozone and OH reactivity in the summer daytime that is not well captured by bulk monoterpene concentration. This reactivity is dominated by limonene, which contributes ∼80% and ∼65%to light dependent sourced ozone and OH reactivity and ∼20% to light independent sourced ozone and OH reactivity. In a changing climate, these BVOC emission sources may vary. For

example, drought may decrease vegetative growth which could increase per-unit-leaf-area in emissions for stored (i.e., light independent) monoterpenes, even as canopy leaf area declines (Lewinsohn et al., 1993; Funk et al., 2004). But, increased precipitation can decrease photosynthesis, causing a decrease in *de novo* (i.e., light dependent) emissions (Lewinsohn et al., 1993; Funk et al., 2004). These findings highlight the need for speciated long-term monitoring studies with a focus on capturing low concentration but highly reactive species.

A significant implication of this work is that the unique drivers of each monoterpene isomer challenge our ability to view this class monolithically or simplify its variability. Measurement studies focused on total BVOC classes may be sufficient to gain an understanding of total BVOC concentrations but demonstrate a need for isomer-resolved understanding of oxidant reactivity. For example, while this work supports the general conclusion that light dependent monoterpenes are a minor component (reflected in current emission models (Guenther et al., 2012) and supported by measurement studies (Bouvier-Brown et al., 2009;

Kesselmeier and Staudt, 1999; Niinemets et al., 2002; Tingey et al., 1979; Davison et al., 2009; Taipale et al., 2011; Rinne et al., 2002), the composition and temporal variability of light dependent monoterpenes, as well as their high per-molecule reactivity, drive strong atmospheric impacts. It is clear that drivers of limonene and sabinene emissions are particularly critical for understanding this ecosystem (see also (McGlynn et al., 2021). Capturing the detail of this or any monoterpene in emissions models is difficult, as the light dependent fraction depends on plant species and other ecological variables, but it is clear

there is some disconnect between the results here and dominant models that, for example, estimate $\alpha$-pinene as more strongly light dependent than limonene (Guenther et al., 2012) and do not tend to vary light dependent fraction by plant function type. Small gaps such as these in our understanding of what drives monoterpene emissions may lead to significant uncertainty in models or outcomes with respect to oxidation and oxidant chemical loss. Furthermore, oxidation of these compounds ultimately leads to SOA formation, but the impacts on this process of the different long- and short-term temporal trends of each

isomer is difficult to assess. It is clear from existing literature that SOA yields vary significantly by isomer and are dependent on structure (Lee et al., 2006; Faiola et al., 2018; Friedman and Farmer, 2018; Lim and Ziemann, 2009). Consequently, we anticipate that light dependent and independent monoterpenes vary in their average SOA yields, and the seasonal and interannual variability observed in this work has significant regional impacts on aerosol loadings. Unfortunately, these differences are difficult to quantify, with previous studies even disagreeing on whether $\alpha$-pinene or limonene has a higher SOA yield (Faiola

et al., 2018; Friedman and Farmer, 2018). Enhanced monitoring of BVOC concentrations and emissions needs to be supple-



mented by improved chemically-resolved measurements of SOA concentrations and formation processes in order to enhance our understanding of the contribution of these emissions to SOA mass loadings.

*Acknowledgements.* This research was funded by the National Science Foundation (AGS 1837882 and AGS 1837891). Tower maintenance and operation was supported in part by the Pace Endowment. D.F.M. and L.E.R.B. were supported in part by Virginia Space Grant Consortium
Graduate Research Fellowships. The authors gratefully acknowledge the assistance of Koong Yi, and Bradley Sutliff in their support in upkeep and maintenance of the instrument at Pace Tower.

*Data availability.* The data set can be found at: https://data.mendeley.com/datasets/jx3vn5xxcn/2

*Author contributions.* D.F.M. analyzed the data, ran the PMF algorithm, produced the figures, and wrote the manuscript. G.F. assisted in deploying and maintenance of the instrument, L.E.R.B. assists in keeping the instrument storage shed running, M.T.L. and S.E.P. contributed
intellectual knowledge and provided feedback on the manuscript, G.I.V. directed the instrument development, research directions, and wrote the manuscript

*Competing interests.* Authors declare no competing interests.





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
