# Peer review of "Minor contributions of daytime monoterpenes are major contributors to atmospheric reactivity"

_Biogeosciences, 2022_

## Author Comment (AC1)

The authors would like to thank the reviewer for their comments and feedback on this manuscript. The reviewer comments have improved the quality and clarity of the research article. The authors have revised the manuscript in accordance with the reviewer comments. Line numbers referred to in the author response are the updated numbers after revision. The reviewer comments are in blue while the author's responses are in **black:**

**Reviewer 1:**

**General comments:**

This paper makes an excellent argument for the need of long-term speciated terpene data to correctly model atmospheric chemistry. Compounds with small ambient concentrations can still play a large role in oxidative capacity and potentially aerosol formation.

To me, this paper provides quantitative support for something that has been stated or implied in the literature for some time. I do not consider the light-dependence emission of limonene or the light-independent emission of a-pinene to be novel, but the quantification of these patterns is nice to see. It gives the atmospheric community a lot to think about in terms of how to approach modeling these complicated systems.

We thank the reviewer for their acknowledgement of the contribution of this work to the field. We hope that our revisions adequately address their comments and concerns.

This paper raises two question in my mind:

Since there is a link between chemical structure and reactivity and some of the light-dependent emissions are highly reactive, then there must be a link between chemical structure and emission pathway (or at least internal synthesis). Can the authors explore this?

The reviewer raises a very interesting observation. We are interested in better understanding these possible connections and are conducting emissions experiments to explore this question. While outside the scope of the present manuscript, we are working to address this question in current measurements.

If compounds with small ambient concentrations matter, what about sesquiterepenes? I can see that they were measured in this study (at least 2 are mentioned), but they are not mentioned in the results. How do they factor into this conversation?

In our previous work, the two measured sesquiterpenes were found to not contribute significantly to calculated atmospheric reactivity. However, the measurement methods used in this work also may not capture all sesquiterpenes. Recent work out of our research group at the same measurement site that more comprehensively targeted sesquiterpenes reached a similar conclusion, that sesquiterpenes do not appear to contribute very much reactivity or aerosol formation potential, though in certain cases they may impact ozone reactivity to some extent (*Frazier et al., 2022*, https://doi.org/10.1039/D2EA00059H). Given this, we do not include them

in this work. We have included a sentence in the manuscript to indicate the reason for their exclusion at line 143-144.

*While the measurement method captures two sesquiterpenes, they are not included in the analysis because these and related measurements have found they do not contribute significantly to most oxidant reactivity (Frazier et al., 2022, McGlynn et al., 2022).*

**Specific Comments:**

The introduction needs some work. First, the references are sometimes distracting; some are repeated multiple times – for example, it is in each sentence within one paragraph. I think there is a more efficient way to use the literature.

We agree with the reviewer's concerns. Repetitive citations have been  removed and the use of excessive citations has been trimmed.

I'm not sure that all references are necessary in each sentence; ensure that they all refer to what you think they do. I do not understand the "discrepancy" found in the literature. Are the authors trying to say there is disagreement about what terpenes are light- or temperature-dependent? I think this entire paragraph should be reframed. The emission of most monoterpenes are thought to be light-independent, but there is some evidence for light-dependent emission (similar to isoprene). This paper is going to use factor analysis to tease out how much of the emissions are in each category.

Thank you for your suggestions. This section of the introduction has been significantly edited (lines 42-54):

*However, some plants do produce and emit monoterpenes in a light-dependent manner (Staudt et al., 1999, Staudt and Seufert, 1995,; Harley et al., 2014, Yu et al., 2017, Taipale et al., 2011; Guenther et al., 2012). Despite these findings, light dependent monoterpene emissions have largely been deemed to contribute minimally to total monoterpene emissions. (Bouvier-Brown et al., 2009, Lerdau et al., 2003). Some studies suggest that this lack of contribution to total flux occurs because they are emitted from only a handful of plant taxa and the emission rates themselves have not been shown to be significant (Staudt et al., 2000, Loreto et al., 1998,  Staudt et al.,1995). However, a few studies find that many trees emit low levels of monoterpenes in a light dependent manner, and that these emissions are seasonal and change with phenological patterns (Fischbach et al., 2002,Ghirardo et al.,2010, Taipale et al.,2011, Steinbrecher et al.,1999}. Overall, understanding of the scale and  seasonality of de novo monoterpene emissions is limited and highly variable in the literature. A major goal of the present work is to understand the potential role that the minor contribution of light dependent emissions and/or individual compounds with differing temporal variability may play in the atmosphere. Certain monoterpenes that are often emitted at low levels and/or in a light dependent manner have extremely high reactivities, raising the question of whether or not chemical impact may be disproportionate to flux magnitude.*

Although this is not the paper that describes the location and instrumentation, I think a little more detail is warranted. Specifically, I would like to see how sampling and calibration was accomplished because you are highlighting reactive species. Accounting for reactivity and deposition is so critical to the accuracy of these measurements.

The authors agree that additional details pertaining to sampling and calibration would benefit the manuscript. The following has been added to the methods section (lines 82-94):

*In brief, air is pulled from mid-canopy (~20 m above ground level) through an insulated and heated Teflon tube. Ozone is removed from the sample using a sodium thiosulfate infused quartz fiber filter (Pollmann et al., 2005) at the front of the inlet. Samples were collected mid-canopy in order to more closely represent the in-canopy environment for co-located studies seeking to understand ozone loss processes. A subsample of air is concentrated onto a multibed adsorbent trap, the details of which can be found in (McGlynn et al., 2021). A custom LabVIEW program (National Instruments) operates the instrument for hourly automated sample collection and analysis. Following sample collection, the trap is thermally desorbed to transfer the sample to the head of the GC column; details pertaining to GC run methods, column, and gas flow rates can be found in (McGlynn et al., 2021}.*

*The instrument is calibrated using a multi-component calibrant (Apel-Riemer Environmental Inc.) optionally mixed at one of four different flows to generate four different mixing ratios. A calibration sample occurs once every seven hours, rotating between zero air only, a calibrant at a fixed "tracking" mixing ratio, and a calibrant at one of three other mixing ratios. Details pertaining to calibrant composition, concentrations, peak integration, and data uncertainty can be found in (McGlynn et al., 2021}.*

In that same vein, Lines 115-116 is confusing – normalized values are multiplied by the sum of the concentrations – how does that provide speciated data?

We recognize that sentence was confusing. It referred simply to how the output of the PMF software needs to be scaled to put it into concentration units using the measured data. To avoid confusion we have removed the sentence and directed readers to the software documentation if interested (line 122):

*Further information on the PMF output can be found in Norris et al., (2014}.*

Limonene is used throughout as a model for light-dependent emissions, but according to Table 1, only 57% of its emissions are described that way. Can you comment on this?

The reviewer is of course correct that limonene has multiple sources, which is apparent in Figure 1, in which limonene exhibits differing diurnal profiles between seasons. Because limonene is the major contributor to LDF monoterpenes, and because it has high reactivity so exerts some outsize influence, we use it throughout the manuscript as an example of the importance of daytime monoterpenes, but did not intend to imply it was a model LDF per se. We have tried to

make that clear in the revisions at a few points throughout the manuscript. Most explicitly, we have added the language below to the discussion of the PMF factors (line 203-206):

*It is important to note that most monoterpenes are split between the two factors and vary within the year, likely because of changing phenological patterns. While some compounds such as α- and β-pinene are almost wholly found in the light independent factor, most of the compounds in the light dependent factor, such as limonene, still exhibit a strong light independent component.*

**Technical comments:**

There are some wording issues. For example, I think line 13 is better communicated as: "the need to monitor species with high atmospheric reactivity, even though they have low concentrations, to more accurately…"

Addressed

Line 18-19: "with secondary effects of other ecological factors" doesn't make sense to me. I understand ecological effects play a role in BVOC emissions, but what is meant by "secondary effects"?

This has been changed to:

"with secondary effects from other factors such as meteorology and deposition"

A few issues with parentheses (missing or incorrectly placed) – e.g. Line 49, 50, 248

Addressed

The formatting is awkward when referring to papers that provided software or equations.

Addressed

Line 99: between May and September

Addressed

Line 179: the black arrows are not present in (b)

Addressed

Figure 3: the caption should have "and" instead of a comma. The black arrows should be noted in the caption.

Addressed

Table 1: the caption is insufficient; it implies that the percentages are calculated based only on light-dependent emissions. LIF an LDF acronyms must be defined here. Why are there only 11 species – where is the data for the other measured species? I think isoprene would be a great addition, since it is referenced and plotted throughout the paper. Also, the sesquiterpene data would be interesting (see comment above).

The caption now reads: *Percent of concentrations attributed to light independent (LIF) and light dependent (LDF) emissions between September 2019 - September 2020 and in summer 2020 (June, July, August).*

The PMF was performed with only monoterpenes to avoid weighting it toward finding an LDF factor; inclusion of isoprene yields similar results, but we felt it was less of an independent result as isoprene strongly pushes the algorithm toward the presence of such a factor. Because this table shows the quantitative split provided by the PMF, there is no way to include the compounds not included in the PMF like isoprene or sesquiterpenes

Line 201: "illuminate's clear differences" – first, the apostrophe is incorrect. Secondly, I don't see clear differences in Figure 5; the shape is actually quite similar.

The grammatical error has been corrected. The authors agree that the wording in this section can be clarified. The section now reads (line 215-218):

*Despite the low contribution of the light dependent factor to total monoterpene concentration, this factor has a large impact on ozone and OH reactivity. Comparing the stacked diurnal concentration profile (Figure 5a) to the stacked ozone and OH reactivity diurnal profile (Figure 5b, c) in summer, limonene and a-pinene prevail as the major contributors to both ozone and OH reactivity.*

Figure 5: explain the dotted lines in the caption. Citing "figure 5a" twice in the same line (214) is not necessary.

Addressed

Word choice should be considered with care. Words like "belies" and "concomitantly" are not commonly used.

We agree, and have changed "belies." However, we feel that the specific meaning of concomitantly very aptly conveys our meaning in this case and have elected to retain it.

Line 233: is the 20% for both ozone and OH reactivity?

Yes

Lines 253-255: This sentence is very awkward and should be rephrased.

The authors appreciate the reviewer for pointing out this run-on sentence. The authors have changed this sentence to (line 264-268):

*Capturing the detail of this or any monoterpene in emissions models is difficult, as the light dependent fraction depends on plant species and other ecological variables. However, it is clear there is some disconnect between the results here and dominant models that, for example, estimate a-pinene as more strongly light dependent than limonene (Guenther et al., 2012) and do not tend to vary light dependent fraction by plant function type.*

---

## Author Comment (AC2)

The authors would like to thank the reviewer for their comments and feedback on this manuscript. The reviewer comments have improved the quality and clarity of the research article. The authors have revised the manuscript in accordance with the reviewer comments. Line numbers referred to in the author response are the updated numbers after revision. The reviewer comments are in blue while the author's responses are in **black:**

**Reviewer 2:**

The manuscript by McGlynn et al. describes measurements of BVOCs in a US forest across two consecutive growing seasons. The authors focus their analysis on the observed monoterpenes and highlight the different behaviour of some monoterpene species with respect to incoming solar radiation. In particular, they describe how the emissions of some monoterpenes appear driven by incoming radiation as opposed to temperature alone, and how in turn these affect the overall reactivity of monoterpenes towards the two main atmospheric oxidants, OH and ozone.

The data presented ultimately supports the conclusions drawn by the authors. However I found some of the vocabulary used in the description of the results somewhat hyperbolic. I also found the method section lacking in detail, in particular the part about the measurement site and the measurement routine (see specific comments below). I understand the authors cite a paper where these details are presented, but the information in that reference should be summarised here to allow this manuscript to stand on its own.

There are also a number of sentences and expressions throughout the manuscript that come across as rather vague and imprecise, and these should be made clearer. Overall, the manuscript would have benefitted from a more thorough proof-read.

In summary, I recommend publication once the comments below are addressed.

**Specific Comments**

Line 11 – "driver" is not the correct word here. You are separating by "emission type" (light dependent vs light independent)

"Driver" was replaced by "emission type".

Line 16 – BVOCs are indeed SOA precursors but they are not precursors to oxidation reactions, they are co-reagents! This needs to be rephrased for clarity.

This sentence has been edited to say (line 16-17):

*Biogenic volatile organic compounds (BVOCs) are important chemical sinks for atmospheric oxidants and precursors for secondary organic aerosol (SOA) and ozone formation.*

Line 21 – "they also require light". This is a bit vague, please rephrase to something along the lines of "they are linked to photosynthesis and therefore require photosynthetically active radiation (PAR)"

This sentence has been rephrased and not reads (line 21-22):

*These emissions tend to increase with temperature (Guenther et al., 2006, Guenther et al., 1997} but are also linked to photosynthesis and therefore require photosynthetically active radiation (PAR).*

Line 37 – also add dispersion as a cause of the drop in concentration

Sentence has been revised to (line 36-37):

*Concentrations of de novo emitted species concomitantly drop as suspended gases are depleted by atmospheric oxidation, deposited to surfaces, and diluted through dispersion.*

Lines 50 and 51 – representation where? I assume the authors mean something along the lines of "Despite accounting for light dependent and independent monoterpene emissions in models, discrepancies exist between these models and observations". This whole sentence needs to be revised as it is unclear.

This section has been reworked and now reads (line 39-54):

*Consequently, monoterpene concentrations are often greatest during the nighttime hours, when oxidation by photochemically formed hydroxyl radicals is minimal and boundary height is reduced, decreasing dilution through atmospheric mixing (Bouvier-Brown et al., 2009; Haa-panala et al., 2007; Panopoulou et al., 2020; Hakola et al., 2012). However, some plants do produce and emit monoterpenes in a light-dependent manner (Staudt et al., 1999; Staudt and Seufert, 1995; Harley et al., 2014; Yu et al., 2017; Taipale et al., 2011; Guenther et al., 2012). Despite these findings, light dependent monoterpene emissions have largely been deemed to contribute minimally to total monoterpene emissions. (Bouvier-Brown et al., 2009; Lerdau and Gray, 2003). Some studies suggest that this lack of contribution to total flux occurs because they are emitted from only a handful of plant taxa and the emission rates themselves have not been shown to be significant (Staudt et al., 1999; Loreto et al., 1998; Staudt and Seufert, 1995). However, a few studies find that many trees emit low levels of monoterpenes in a light dependent manner, and that these emissions are seasonal and change with phenological patterns (Fischbach et al., 2002; Ghirardo et al., 2010; Taipale et al., 2011; Steinbrecher et al., 1999). Overall, understanding of the scale and seasonality of de novo monoterpene emissions is limited and highly variable in the literature, making accurate representation in emission and chemical models difficult. A major goal of the present work is to understand the potential role that the minor contribution of light dependent emissions and/or individual compounds with differing temporal variability may play in the atmosphere. Certain monoterpenes that are often emitted at low levels and/or in a light dependent manner have extremely high reactivities, raising the question of whether or not chemical impact may be disproportionate to flux magnitude.*

Line 77 – you need to also add OH reactivity

Fixed

Fixed

Line 88 – The authors need to provide more details on the measurement site. What type of forest is it? What are the dominant emitters of monoterpenes?

This information has been added to the methods section (line 77-80):

*The forest is largely composed of oak, maple, and pine trees; oak predominantly emits isoprene while pine is a major source of monoterpenes and sesquiterpenes. Additional information pertaining to the measurement site can be found in McGlynn et al., (2021).*

Line 91 – Can the authors comment on the choice of sampling air from mid-canopy rather than from the top of the canopy? Any advantages, disadvantages or caveats?

Samples were collected from mid-canopy in order to more closely represent the in-canopy environment. By minimizing transport between the emissions sources (leaves) and the sampling inlet, we sought to mitigate potential reactive losses prior to reaching the sample system. Associated research was focused on understanding ozone within the canopy, and these measurements help constrain chemical loss. Sampling above the canopy would provide useful information on emissions if we had a gradient measurement, but the time resolution of this instrument precludes that sampling strategy (we are working on a related two-channel instrument to make that work). These data represent in-canopy concentrations and atmospheric composition but consequently provide a more limited understanding of the impacts of these emissions more regionally, since reactive loss could occur before being emitted from the canopy and advected.

We have added a sentence paraphrasing the reasoning for this approach in the methods section (line 84-85):

*Samples were collected mid-canopy in order to more closely represent the in-canopy environment for co-located studies seeking to understand ozone loss processes.*

Lines 92-95 – Please provide an brief account of how the GC-FID was calibrated.

This information has been added to the methods section (line 90-94):

*The instrument is calibrated using a multi-component calibrant (Apel-Riemer Environmental Inc.) optionally mixed at one of four different flows to generate four different mixing ratios. A calibration sample occurs once every seven hours, rotating between zero air only, a calibrant at a fixed "tracking" mixing ratio, and a calibrant at one of three other mixing ratios. Details pertaining to calibrant composition, concentrations, peak integration, and data uncertainty can be found in (McGlynn et al., 2021}.*

Line 112 (equation (1)) – please explain where the value of 0.15 arises from

This value is recommended by Norris et al. as an estimate of overall uncertainty in the data. It reasonably represents the uncertainty of this instrument as well based on uncertainties in calibration slopes, and inherent uncertainty in integration of chromatographic peaks, which has been shown to be on the order of 10-15% (Isaacman-VanWertz et al., 2017, https://www.doi.org/10.1016/j.chroma.2017.11.005).

This has been added to the manuscript (line 119-122):

*The uncertainty value of 0.15 is recommended by Norris et al.( 2014) as an estimate of overall uncertainty in the data. It reasonably represents the uncertainty of this instrument as well based on uncertainties in calibration slopes, and inherent uncertainty in integration of chromatographic peaks, which has been shown to be on the order of 10-15% (Isaacman-VanWertz et al., 2017).*

Line 127 – replace ~10% with ~10-16%. You gave a range on line 125, it needs to be double that range here.

Fixed

Line 128 – Given how some of these reaction coefficients are poorly characterised, what additional uncertainty does this add to the reactivity calculations, and ultimately your conclusions? Can the authors comment on this?

The reviewer is correct that there could be additional, unaccounted for uncertainty arising from the reaction rate constants. *Some rate constants such as thujene were calculated from structure activity relationships and previous work has found that calculated rate constants add significant uncertainty to calculated ozone reactivity (Frazier et al. 2022). However, compounds that contribute the most to atmospheric reactivity, such as a-pinene, limonene, and sabinene have measured rate constants, therefore, we do not expect significant uncertainty in our calculations.* We have added this comment to the manuscript (line 134-137).

Line 141 – Emissions are temperature driven, not concentrations!

This has been changed to: "light-independent emitted monoterpenes"

Figure 3 – There appear to be some very low values in all three panels after Jul 2020 and before the gap in the timeseries. Are these real? If not, please remove.

These values have been removed.

Caption to Figure 3 – Please state what the three arrows indicate. I know it is in the main text, but the caption and the figure it accompanies need to be somewhat "self-contained"

This sentence has been added to the caption to address this comment:

*The black arrows in figure 3a denote the transient periods that are apparent in both the isoprene data (3a) and the light dependent factor (3b).*

Figure 4 – A correlation plot of the Light Dependent and Light Independent monoterpenes vs Isoprene would support what the authors say in lines 186-187 better than these three timeseries.

We agree that a correlation plot between isoprene vs. LDF and isoprene vs. LIF would support the assertion that some compounds vary in a light dependent manner. However, we feel that the overview in Figure 3 and the snapshot in Figure 4 help illustrate the correlated seasonality and diurnality more clearly than a scatter plot. We have consequently included the scatter plot as a supporting figure in the SI and reference in the text as described below.

[Figure]

Figure S3. Correlation plots of isoprene with (a) the light dependent factor (LDF) and (b) the light independent factor (LIF) during the summer of 2020. Linear regression equations and $r^2$ is provided on each plot.

In the results section of the main text, we added (line 199-201):

*Additionally, isoprene concentrations correlates reasonably well with light dependent monoterpenes during summer ($r^2$=0.57, Figure S3a) and does not correlate with light independent monoterpenes ($r^2$=0.01, Figure S3b).*

Line 204 and then again Line 212 – I have an issue with the words "prevail" and "enormous" in this context. I can see the peak emergence in the reactivity plots in Figure 5, but it still does not prevail over the night-time peak. Please rephrase. Also replace "enormous" with "significant"

Addressed

Figure 5 – Why not show both years together?

Because the two years were processed independently, they cannot be readily combined into the same set of sub-figures. Consequently, including the second year would require including a second analogous set of sub-figures, which we felt would be unwieldy considering the results are very similar and the findings could be adequately conveyed with just one year. A few lines have been added to this section of the manuscript to indicate that the second year can be found in the supplemental document (line 222-223):

*PMF results from 2020-2021 are generally very similar to the results shown here in terms of diurnality and composition (Fig. S5).*

Caption to Figure 5 – Please explain what the dashed lines are

This has been added to the caption of Figure 5:

*The dashed lines in a, b, and c represent the contribution from LD monoterpenes (d, e and f) while the dotted lines represent the contribution from LI monoterpenes (g, h, and i).*

Line 214 – It is not that they "have little diurnal pattern", but they have a less pronounced one. Please rephrase.

Addressed

Line 226 – Can the authors speculate on what processes might lead to the emission of LD monoterpenes?

We are very interested in better understanding what processes might lead to LD emissions of monoterpenes and are conducting emissions experiments to explore this question. While outside the scope of the present manuscript, we are currently conducting research to explore this question.

**Technical Corrections**

Line 9 – should read "concentrations" (plural)

Fixed

Line 7 – add "USA" after Virginia

Fixed

Line 24 – "primarily with a temperature dependence". A bit colloquial. Rephrase to "primarily driven by temperature".

This sentence has been edited to (line 24-25):

In contrast, other emissions occur independently of light and are driven by temperature from a wide variety of vegetation, and therefore occur year-round.

Line 34 – should read "peak around midday"

Fixed

Line 35 – replace light with PAR

Fixed

Line 55 – replace "not light dependent" with "independent of light"

Fixed

Line 79 – replace "coming" with "arising"

Fixed

Line 86 – Virginial should read Virginia

Fixed

Line 88 – add USA after VA.

Fixed

Line 89 – replace "a gas chromatography flame ionization detector" with "a gas chromatograph with flame ionization detection"

Fixed

Line 91 – replace "seen" with "found"

This has been reworded due to edits made to address previous reviewer comments.

Line 118 – "i" should be subscripted

Fixed

Line 121 – add "are": "are listed"

Fixed

Line 132 – add "to" after "contribute", and replace "large" with "largest"

The grammatical suggestion has been taken into consideration and reworded accordingly.

Line 140 – replace "evening" with "night-time"

All instances of evening have been replaced with night-time.

Line 141 – replace "due to" with "modulated by"

Fixed

Line 148 – replace "lows… highs" with "minima… maxima"

Fixed

Line 150 – "the reaction rate of limonene with OH radical, and ozone, is 3, and 2.3, times as fast, respectively, as those of α-pinene." Very convoluted with all the commas. Replace with "the reaction rates of limonene with the OH radical and ozone are, respectively, 3 and 2.3 times faster than those of α-pinene"

Fixed

Line 158 – add "to" after "through"

We do not see the error to which the reviewer refers and have not made any change.

Line 164 – "Driver" should be plural

Fixed

Line 165 – remove "species", it is redundant

Fixed

Lines 171-172 – Replace "exhibit a tendency to have high daytime concentrations" with "exhibit a tendency to daytime peaks".

Changed to "exhibit a tendency toward daytime maxima"

Line 195 – "more evenly split"?

Fixed

Line 201 – I think the authors mean "illustrates" here?

This has been changed to illuminates. The previous grammatical error has been corrected.

Line 205 – add "relatively" before "lower"

This has been changed to relatively low.

Line 252 – Replace "Small gaps such as these" with "These small gaps"

Fixed